# Different Gut Microbiome Profiles in Patients with Transthyretin Amyloidosis with and Without Cardiac Involvement

**DOI:** 10.3390/ijms26041689

**Published:** 2025-02-16

**Authors:** João Henrique Rissato, Natalia de Melo Pereira, Cristhian Espinoza Romero, Georgina del Cisne Jadán Luzuriaga, Bruno Vaz Kerges Bueno, Caio Rebouças Fonseca Cafezeiro, Aristóteles Comte de Alencar Neto, Thaís Sousa Borges, Suenia Freitas Carvalhal, Félix Jose Alvarez Ramires, Luciano Nastari, Charles Mady, Fábio Fernandes

**Affiliations:** Heart Institute (InCor) do Hospital das Clínicas da Faculdade de Medicina da Universidade de São Paulo, São Paulo 05508-220, Brazil; n.melo1@hc.fm.usp.br (N.d.M.P.); cristhian.espinoza@hc.fm.usp.br (C.E.R.); geojadan@gmail.com (G.d.C.J.L.); vazkerges@yahoo.com.br (B.V.K.B.); caiocafezeiro@hotmail.com (C.R.F.C.); aristotelesalencar@gmail.com (A.C.d.A.N.); thais.borges@hc.fm.usp.br (T.S.B.); suenia.carvalhal@hc.fm.usp.br (S.F.C.); felix.ramires@incor.usp.br (F.J.A.R.); luciano-nastari@uol.com.br (L.N.); charles.mady@incor.usp.br (C.M.); fabio.fernandes@incor.usp.br (F.F.)

**Keywords:** amyloidosis, gastrointestinal microbiome, heart failure, cardiomyopathies, gastrointestinal tract

## Abstract

Transthyretin amyloidosis (ATTR amyloidosis) is characterized by the buildup of amyloid protein in organs like the gut and the heart. As a result, hypoperfusion, edema, and dysautonomia cause an imbalance in the gut microbiome. We aimed to identify the gut microbiome composition in ATTR amyloidosis patients with and without heart involvement, as well in controls. Sixty participants were divided into three groups: 20 with ATTR amyloidosis and heart involvement (G1), 19 with ATTR amyloidosis but no heart disease (G2), and 21 controls (G3). The microbiome profiles were obtained through 16S rRNA gene sequencing. Additional evaluations included a clinical questionnaire, echocardiogram, six-minute walk tests, troponin, BNP, and genotype analysis. Compared to G3, G1, and G2 groups had different levels of *Streptococcus*, *Lachnospiraceae*, and *Sellimonas*, while the controls showed a higher relative abundance of *Methanosphaera*. *Streptococcus* was linked to higher troponin levels. *Lachnospiraceae* was associated with lower BNP levels and smaller left atrium volumes. *Sellimonas* was associated with a higher intestinal symptom score, while *Methanosphaera* with a lower symptom score. ATTR amyloidosis patients have a different intestinal microbiome profile compared to the control group. There were correlations with genotype, gastrointestinal symptoms, heart failure biomarkers, echocardiographic parameters, and the six-minute walk test.

## 1. Introduction

ATTR amyloidosis is a systemic disease characterized by the buildup of amyloid protein in various organs and tissues. Amyloid transthyretin (ATTR) is mainly produced and released into the bloodstream by the liver. The culmination of the disease’s pathophysiology is the degeneration of the protein tetramer and its deposition in the tissues [1], leading to renal failure, erectile dysfunction, carpal tunnel syndrome, vitreous opacity, glaucoma, as well as in the gut, causing diarrhea, constipation, nausea, and weight loss. It can also manifest itself as a polyneuropathy [2]. In the heart, this process causes toxicity, thickening, and varying degrees of diastolic and systolic dysfunction.

Heart failure (HF) provokes systemic tissue hypoperfusion by the low cardiac output and vasoconstriction caused by neuroendocrine activation. This results in hypoxia, leading to intracellular acidosis in the intestinal villi [3]. A sodium (Na) pump becomes hyperactivated to correct intracellular acidosis, causing H^+^ ions to be moved into the intestinal lumen while more Na ions are absorbed. Additionally, due to the increased preload in HF, the intestinal villi are more likely to swell, and the enterocytes may not receive enough trophic factors. It weakens the intestinal barrier and makes it easier for bacteria to move from the gut into the bloodstream, as portrayed by the higher levels of immunoglobulin A (IgA) anti-lipopolysaccharide (LPS) found in advanced HF patients. This sets off a vicious cycle of worsening HF by fluid retention, resulting in sodium being internalized. The change in intestinal pH, now tending towards acidification, promotes the growth of certain groups of microorganisms at the expense of other bacterial populations, leading to an increase in the production of harmful metabolites to the circulatory system, such as trimethylamine oxidase (TMAO), and a decrease in the production of short-chain fatty acids like butyrate, which is a nourishing substance for the enterocyte.

### The Gut Microbiome

The human gut microbiome is incredibly diverse [4]. It is unique to each individual and shaped by genetic factors, age, type of delivery, antibiotic use, and diet [5]. These microorganisms have been found to play a crucial role in food digestion, vitamin synthesis, production of short-chain fatty acids, immune system modulation, and protection against infections. The term “microbiota” refers to the group of microorganisms living in a particular biome or habitat. The microbiome refers to the genetic material of these microorganisms. Recent data suggests that the resilience of the intestinal microbiota may play a role in the development of diseases such as obesity, atherosclerosis, and systemic arterial hypertension [6].

The hypothesis is that there must be taxonomic patterns in the intestinal microbiome in ATTR amyloidosis, given the alterations in the intestinal microbiome may be due not only to disorders related to the digestive tract (such as dysautonomia and deposits in the intestinal tissue itself) but also to reduced blood flow and intestinal swelling as a result of heart failure as the disease advances (Figure 1).

In this study, we aimed to identify the composition of the intestinal microbiome in three groups of patients and correlate the taxonomic groups with gastrointestinal clinical symptoms score, genotype, echocardiogram data, laboratory markers (type B natriuretic peptide (BNP) and troponin I), and a six-minute walk test.

Group 1 (G1)—ATTR amyloidosis patients with cardiac involvementGroup 2 (G2)—ATTR amyloidosis patients without cardiac involvementGroup 3 (G3)—Control group with similar living and eating habits to the patient groups

## 2. Results

The average age was 58 years. More than half (51.66%) were male. Of the 39 patients in groups G1 and G2, 15% had the ATTRwt, while the majority (85%) had the ATTRv. Among those with the hereditary form, the most common mutation was ATTRV142I (52.9%), followed by ATTRV30M (17.6%). The remaining altered genotypes accounted for 26.4% of the patients. Table 1 presents the patients’ clinical characteristics, laboratory data, and echocardiographic features.

### 2.1. Alpha Diversity

The Chao and Shannon indices were higher in the control group. However, when the Kruskal-Wallis test was applied, no significant difference was demonstrated between the three groups (Figure 2A).

### 2.2. Beta Diversity

The results displayed on the Non-Metric Multidimensional Scaling (NMDS) graph show an absence of distinct grouping, with a noticeable overlap of the three groups’ ellipses. Based on this diversity parameter, the statistical test also confirmed the lack of distinction between the groups (Figure 2B).

### 2.3. Relative Abundance and Taxonomic Composition

The bacterial composition did not vary significantly at the phylum, class, order, and family levels. Firmicutes was the most abundant phylum in most samples from all three groups.

Groups 1 and 2 exhibited a differential abundance of the bacterial genera *Streptococcus*, *Lachnospiraceae* UCG-003, and *Sellimonas* compared to the control group (G3), while G3 showed a higher relative abundance of *Methanosphaera* (Figure 3).

We did not find a central microbiome (core microbiome) for either the healthy or sick groups. However, we noticed that 65% of all samples (G1 + G2 + G3) contained the same ASV (Amplicon sequence variant) belonging to the genus *Dorea*.

### 2.4. Correlation with Complementary Exams

#### 2.4.1. Clinical Questionnaire Correlations

The genera *Einsenbergiella* and *Hungatella* were more prevalent among patients with more severe symptoms. Conversely, *Methanosphaera* was more prevalent among those with milder symptoms (Figure 4A). Increasing values in the X-axis correlate with increasing abundance of the different sorts of bacteria displayed.

When addressing symptoms such as alternating bowel movements and constipation (questions 9 to 15), a higher abundance of the genus *Hungatella* was observed, along with the genus *Sellimonas*. Similar to the overall score, the score specific to the lower digestive tract also revealed a higher abundance of the *Methanosphaera* taxon in those experiencing fewer symptoms (Figure 4B).

From this point onwards, the results will only pertain to patients in groups G1 and G2.

#### 2.4.2. Laboratory Markers: BNP and Troponin

Patients with higher levels of troponin I showed a greater abundance of the genera *Intestinomonas*, *Butyricicoccaceae*, and *Streptococcus*. On the other hand, participants with higher circulating levels of BNP were associated with the genus *Hungatella*, while *Lachnospiraceae* UCG-003 was linked to patients with lower levels of BNP (Figure 4D).

#### 2.4.3. Echocardiographic Parameters

Patients with smaller atrial volumes have a higher relative abundance of *Lachnospiraceae* UCG-003, while those with larger atria have a higher relative abundance of *Lachnospiraceae* NK4B4 (Figure 5A). Patients with a higher cardiac mass index show a higher relative abundance of *Butyricicoccaceae* (Figure 5B).

#### 2.4.4. Genotype

Patients with an altered genetic test (ATTRv) showed Instestinimonas as the most abundant genus (Figure 5C). Additionally, *Lachnospiraceae* NK4B4, more prevalent in patients with larger atrial volumes, was also more frequent in patients with the ATTRwt.

#### 2.4.5. 6 Min Walk Test

*Intestinimonas* is the most abundant genus in patients who walked shorter distances during the 6-min walk test. Consequently, this clade was associated with individuals exhibiting the poorest physical performance (Figure 5D).

## 3. Discussion

The main findings of this study focus on the presentation of a microbiome profile distinct from that of the control group. Furthermore, correlations were observed between this profile and genotype, gastrointestinal symptoms, heart failure biomarkers, echocardiographic parameters, and six-minute walk test results.

The gut microbiome’s alpha and beta diversity portrays this microbial ecosystem’s complexity and variability. These measurements provide valuable data on gastrointestinal health and can help guide therapeutic interventions such as microbial modulation through probiotics, prebiotics, and fecal transplantation [7]. 

Our results showed that the Chao [8] and Shannon [9] alpha diversity indices were higher in the control group. Even though the statistical test yielded *p*-values greater than 0.05, it was qualitatively observed that control individuals had greater bacterial diversity, aligning with existing literature data [10].

The taxonomic composition did not clearly separate samples from control and sick individuals when looking at phylum, class, order, and family. In all three groups, the most abundant phylum was Firmicutes, which supports previous studies indicating that the fecal microbiome of mammals is mainly composed of Bacteroidetes and Firmicutes [11]. 

Considering that individuals in the control group had similar eating habits to the sick patients, we hypothesize that dietary patterns may play a more significant role than amyloidosis in shaping the intestinal microbiome from phylum to family levels. This is in line with previous research findings [12].

The search for a core microbiome has been widely used to study taxonomically defined microbial communities, identifying groups of bacteria that are especially common in host populations [13]. The core microbiome generally refers to any group of bacterial taxa typical of the host. Despite the term’s popularity and increasing use, there is little consensus on how to quantify the core microbiome [14].

In this study, we explored the concept of the core microbiome at the level of amplicon sequence variants (ASVs) in two scenarios: ASVs are present in all individuals in a group, and ASVs are present in 65% of the individuals in the group. We found no core microbiome for G1, G2, G1 + G2, and G3, but we identified a core microbiome comprising 65% of all ASVs belonging to the genus *Dorea*.

With increased sample scrutiny, we could detect differences in the abundance of bacterial genera between the patient groups and the control group. Specifically, Groups 1 and 2 exhibited differentially abundant bacterial genera, such as *Streptococcus*, *Lachnospiraceae* UCG-003, and *Sellimonas*, compared to the control group (G3). On the other hand, G3 showed a higher relative abundance of *Methanosphaera*.

Once groups 1 and 2 (with and without cardiac involvement) share similar results of bacteria abundance, it is reasonable to hypothesize that, in this study, the burden of ATTR amyloidosis prevails over heart failure’s influence on the gut microbiome.

Due to the rarity of the disease, there was a shortage of participants, it was not possible to exclude those with associated diabetes for analysis despite the potential influence of this disease on the intestinal microbiome.

It is important to note that if someone experienced persistent diarrhea for over two weeks, they were not included in the study. Also, the three groups of patients had no significant differences in intestinal symptoms or body mass index during the sample collection period (*p* > 0.05). Therefore, the variances we found in this study likely occur in the early stages. This could provide an opportunity for future research into new diagnostic tools for detecting early signs of intestinal involvement in amyloidosis.

### Correlation Between the Most Abundant Taxa and Complementary Assessments

The connection between *Streptococcus* and heart disease has been extensively researched, particularly rheumatic fever and infective endocarditis [15]. For instance, *S. viridans* is one of the primary pathogens associated with this condition, especially in patients with pre-existing heart disease or valve prostheses [16]. *S. bovis* is known to be linked to colon cancer and endocarditis [17]. This study found that the genus *Streptococcus* was associated with a group of sick patients with and without heart disease (G1 and G2) and high troponin levels.

*Sellimonas* is a type of bacteria in the Firmicutes phylum. In a study exploring the causal relationship between intestinal microbiota and endocarditis, it was found that the *Sellimonas* genus, along with other microorganisms, plays a protective role against endocarditis [18]. The genus was also more abundant in patients with more intense lower gastrointestinal symptoms, such as changes in bowel movement frequency and constipation.

The *Methanosphaera* genus is a group of methanogenic archaea found in the human and animal intestines. The exact relationship between *Methanosphaera* and cardiovascular disease is not yet fully understood. Some research suggests that methanogenic archaeal species, such as *M. stadtmanae*, may play a role in the development of systemic diseases such as obesity and intestinal cancer [19]. In our study, the *Methanosphaera* genus was found to be more abundant in control participants (G3), and consistently, it was linked to a lower score of gastrointestinal symptoms.

The *Intestinomonas* genus showed a higher relative abundance in patients with higher troponin levels, poorer walking test performance, and altered genotype. This result makes sense, as the most prevalent genotypic presentation in the studied population of patients with the ATTRv was ATTRV142I (52.9%), which commonly leads to heart disease [19].

*Hungatella* was more abundantly present in patients with higher BNP levels and more intense intestinal symptoms, which is quite plausible considering the varying degrees of edema and hyperperfusion that the intestines experience in the context of HF. *Butyricocchaceae* was found to be more associated with patients with higher troponin levels and cardiac mass index. Interestingly, *Lachnospiraceae* UCG003, associated with patients with cardiac involvement (G1), was paradoxically more related to lower BNP levels and smaller left atrium volumes.

Limitations: There was a significant difference in the distribution of male and female participants between patient groups (1 and 2) and the control group (3). More wives were randomly recruited as control participants with the same dietary pattern as the patients because 15% of patients had ATTRwt, in which the male sex predominates [20]. Likewise, ATTRwt patients, who used to be elderly and suffered mainly from cardiomyopathy, were spontaneously allocated to group 1, contributing to the distribution imbalance by age and genotype among the groups.

Many patients in group 2 will probably still present cardiac involvement during the course of the disease. It would be interesting to analyze the intestinal microbiome in a longitudinal cohort of these patients to observe dynamic changes that could provide additional information on the development of heart disease. 

A further approach to nutritional questions, gut allergies, lifestyle topics such as smoking, and physical pursuits would probably fill some of the present study’s gaps.

Also, due to budget issues, other features that could strengthen the analysis, such as bacteria metabolites and glucose and lipid profiles, were not measured.

The conclusions of this study are based on data that allow associations to be established. Additional methodologies like shotgun metagenomics could help establish causal links between ATTR amyloidosis and the intestinal microbiome. 

## 4. Material and Methods

### 4.1. Study Population

This cross-sectional study included 60 patients of both sexes, aged between 18 and 90 years. The patients were undergoing follow-up at the outpatient clinic. There were three groups: 20 participants in G1, 19 in G2, and 21 in G3. The inclusion of patients took place over 12 months.

### 4.2. Inclusion Criteria

Diagnosis of wild-type transthyretin amyloidosis (ATTRwt) and patients with a hereditary form (ATTRv) documented by a pathogenic transthyretin mutationAll the following criteria have been met to define cardiac involvement by ATTR amyloidosis: evidence of cardiac thickening by echocardiography or magnetic resonance imaging with measurement of the interventricular septum at end-diastole (SIVd) > 12 mm; myocardial scintigraphy with technetium pyrophosphate indicating levels 2 or 3 of uptake in the absence of circulating immunoglobulins; in the presence of uncertain significant hypergammaglobulinemia (MGUS), confirmation of the ATTR in the tissue through immunohistochemistry (IHC) or mass spectrometry was necessary

### 4.3. Exclusion Criteria

Individuals with a history of inflammatory bowel diseaseThose experiencing persistent diarrhea for more than two weeks in the two months before fecal collectionIndividuals who have used antibiotics, prebiotics, and probiotic supplements in the two months before collection

The participants were instructed to wash their hands and use an appropriate collector for defecation, avoiding possible contamination with surfaces and toilet water. Samples were collected at the participants’ homes, preserved in a solution with guanidine, and placed in a closed container under refrigeration (6 to 7 °C) until delivered to the clinical research unit for up to five days. There, they were stored at −20 °C until analysis. 

### 4.4. Automatic Sequencing and Bioinformatics Analysis

The microbiome was characterized by amplifying the V3 and V4 domains of the bacterial 16S ribosomal segment and using the MiSeq Sequencing System from Illumina Inc. (San Diego, CA, USA) for high-throughput sequencing. Alpha diversity was assessed using the Chao1 and Shannon indices with the q2-diversity plugin, and the sequence quality was evaluated with FastQC. QIIME2 v.2022.2, including Dada 2 tool for sequence variants and taxonomic classification with the Silva v138 database. Finally, R v4.1.0 software and RStudio 3.6.0 generated taxonomic composition and diversity figures.

### 4.5. Transthoracic Echocardiogram

The complete resting echocardiographic study was performed using Vivid E9 equipment (GE Healthcare, Milwaukee, WI, USA). More than one examiner obtained all measurements in accordance with the recommendations of the American Society of Echocardiography (ASE), allowing variability of up to 10%. The images were stored in digital format for later analysis.

### 4.6. Complementary Laboratory Tests

BNP and troponin I plasma concentrations were measured using sandwich immunoassay with direct chemiluminescence. The ADVIA Centaur® commercial kit from Siemens Medical Solutions Diagnostic (Munich, Germany) was used to evaluate BNP, while the ADVIA Centaur® TnI-Ultra commercial kit from Siemens Healthcare Diagnostics (Munich, Germany) was used for troponin I. Both samples were analyzed using automated equipment from the same brand.

### 4.7. 6 Min Walk Test

The test was conducted in a 30-m corridor to evaluate the distance covered. 

### 4.8. Clinical Questionnaire

The severity of gastrointestinal symptoms was evaluated using a clinical questionnaire, the Gastrointestinal Symptom Rating Scale (GSRS), validated in Portuguese [21]. It consists of 15 questions, with the first eight focusing on symptoms associated with the upper digestive tract and the remaining seven on symptoms related to the lower digestive tract.

### 4.9. Statistical Analysis

Statistical analyses for comparing alpha diversity were conducted using the Kruskal-Wallis test, and the Permanova test was employed for beta diversity. The DESeq2 method was chosen for the differential abundance analysis, initially developed for RNA-Seq differential expression analysis and validated for metagenomics analysis. An adjusted *p*-value < 0.05 criterion was used to identify differentially abundant taxa, with the correction of standard DESeq2 multiple tests (FDR). Only taxa detected in at least five samples were included in the analysis.

## 5. Conclusions

Patients with ATTR amyloidosis showed a different microbiome profile than those in the control group. The microbiome profile is correlated with genotype, gastrointestinal symptoms, heart failure biomarkers, echocardiographic parameters, and six-minute walk tests.

## Figures and Tables

**Figure 1 ijms-26-01689-f001:**
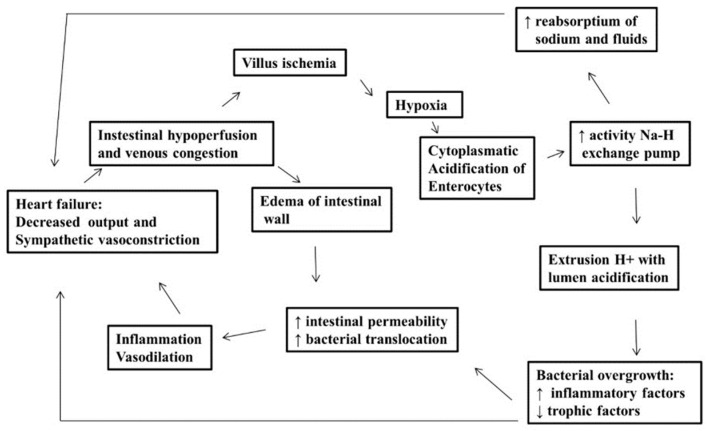
The interface between heart failure and the intestinal microbiome.

**Figure 2 ijms-26-01689-f002:**
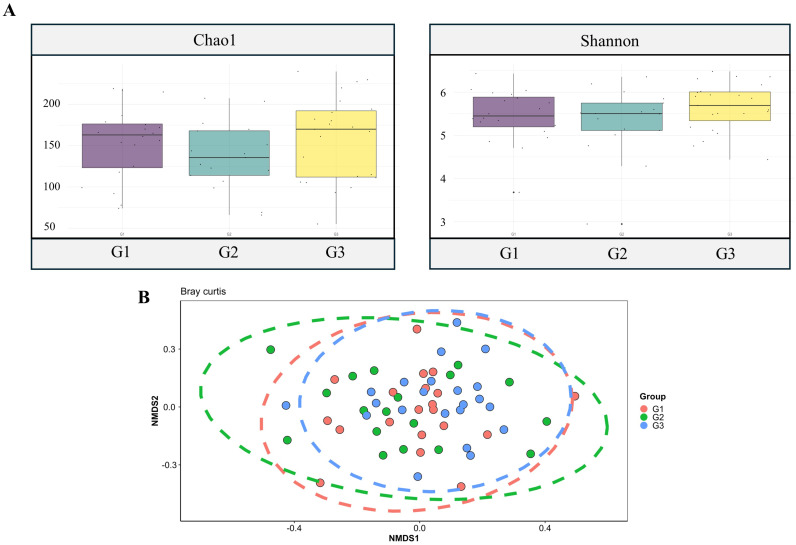
(**A**) Comparison of groups based on alpha diversity. (**B**) Group comparison based on beta diversity using the Bray-Curtis index. Each dot represents a participant.

**Figure 3 ijms-26-01689-f003:**
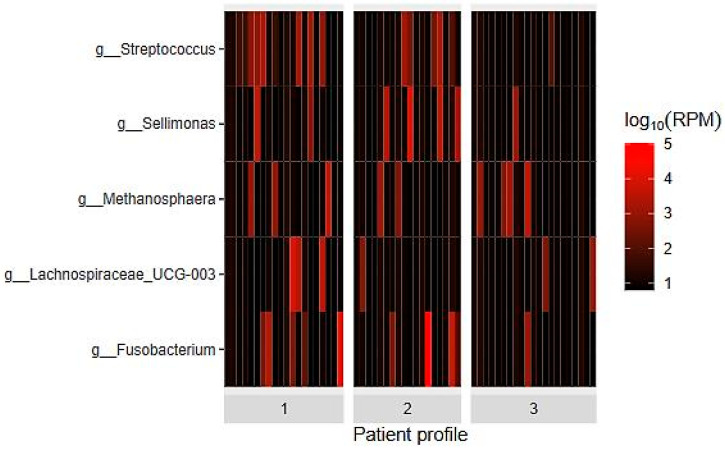
Relative abundance of bacterial genera in the three groups of patients studied (heatmap).

**Figure 4 ijms-26-01689-f004:**
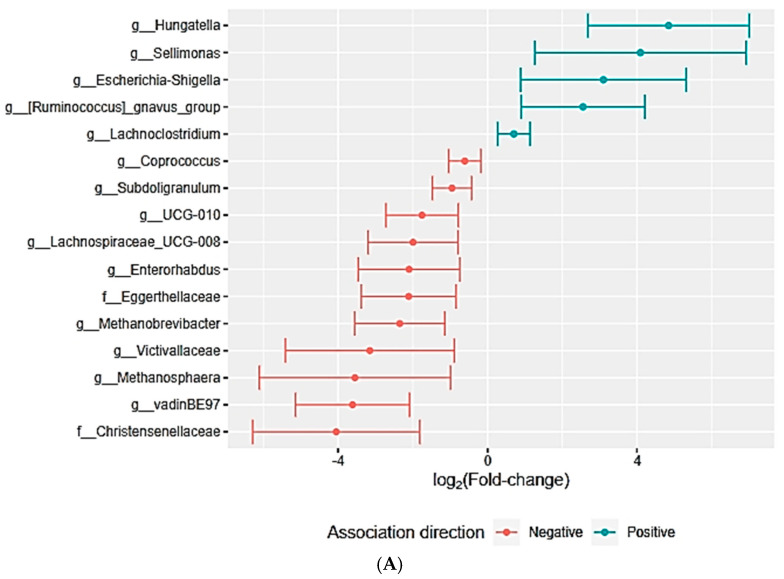
(**A**,**B**) Correlation between general gastrointestinal symptoms and relative abundance of bacterial taxa. (**C**) Correlation between troponin I levels and relative abundance of bacterial taxa. (**D**) Correlation between BNP levels and relative abundance of bacterial taxa.

**Figure 5 ijms-26-01689-f005:**
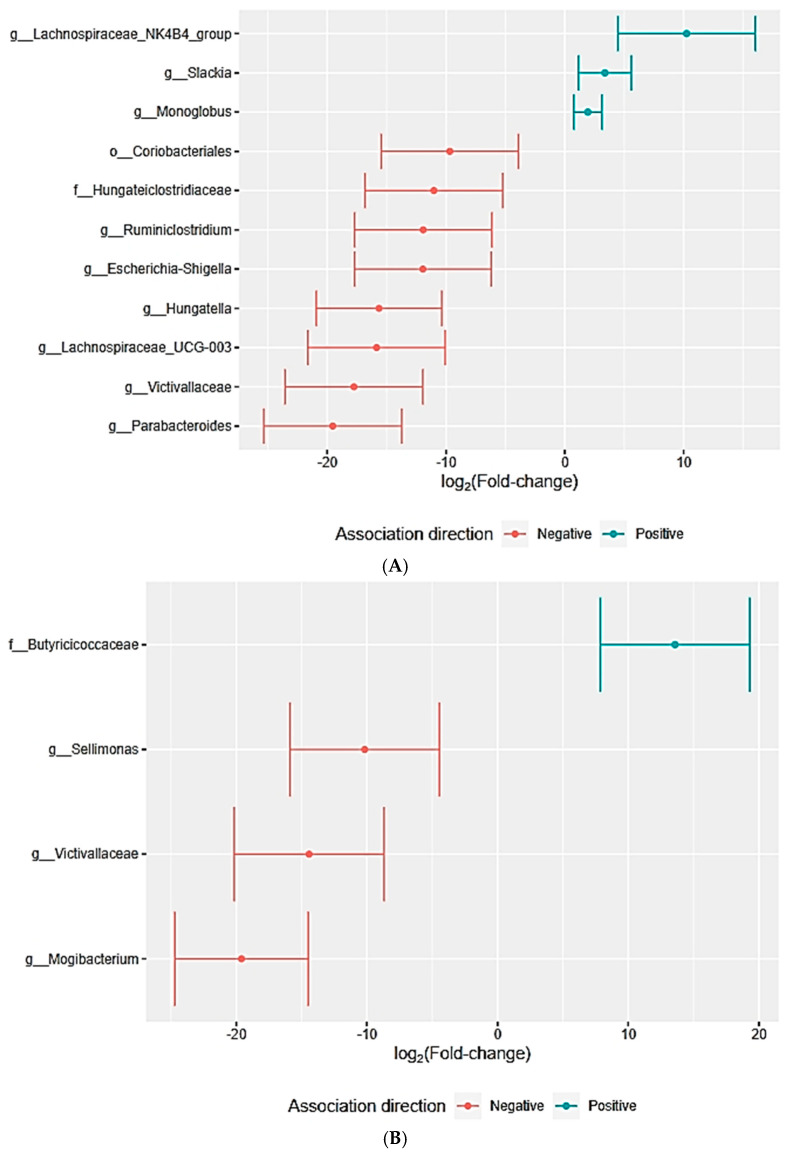
(**A**) Correlation between left atrium volume and relative abundance of bacterial taxa. (**B**) Correlation between cardiac mass index and relative abundance of bacterial taxa. (**C**) Correlation between patient genotype and relative abundance of bacterial genera. (**D**) Correlation between 6MWT and relative abundance of bacterial genera.

**Table 1 ijms-26-01689-t001:** Clinical characteristics of patients, laboratory data, and echocardiographic features of patients.

Clinical Characteristics of Patients *				
	Group 1 (n = 20)	Group 2 (n = 19)	Group 3 (n = 21)	*p* Value
Age (year)	73.8 ± 9.1	42.58 ± 10.87	55.26 ± 20.68	<0.05
Sex (M/F)	17/3	12/7	3/18	<0.05
Systolic blood pressure (mmHg)	119.33 ± 18.14	128.26 ± 17.95	124.71 ± 10.62	0.210
Diastolic blood pressure (mmHg)	74.52 ± 12.67	84.32 ± 11.23	77.19 ± 6.8	0.010
Mean blood pressure (mmHg)	89.14 ± 13.63	98.58 ± 12.14	92.71 ± 6.81	0.030
Heart rate (bpm)	72.67 ± 11.93	70.42 ± 9.86	74.05 ± 10.46	0.570
Oximetry (%)	96.19 ± 1.99	97.47 ± 1.07	97.38 ± 0.92	0.008
Weight (kg)	67.14 ± 10.18	73.84 ± 15.38	70.33 ± 10.05	0.220
Height (cm)	165.76 ± 5.44	168.42 ± 7.77	163.29 ± 6.73	0.060
Body mass index (kg/m^2^)	24.4 ± 3.1	25.67 ± 4.91	26.37 ± 3.14	0.240
Clinical questionnaire score	26.48 ± 8.66	24.74 ± 9.60	25.62 ± 7.3	0.810
Walking test distance 6 min (M)	355.52 ± 77.28	422.23 ± 115.65		0.589
Hypertension	8 (40%)	2 (10.53%)	3 (14.89%)	0.097
Diabetes	8 (40%)	0	4 (19.05%)	0.008
Dyslipidemia	7 (35%)	1 (5.26%)	4 (19.05%)	0.067
	**Group 1 (n = 20)**	**Group 2 (n = 19)**		***p* Value**
BNP (pg/mL)	484.26 ± 309.42	21.37 ± 20.76		0.033
Troponin (ng/L)	46.93 ± 45.39	7.42 ± 10.95		0.228
**Echocardiographic characteristics of patients ****				
	**Group 1 (n = 20)**	**Group 2 (n = 19)**		***p* Value**
LVEF (%) ^1^	45.05 ± 12.26	59.16 ± 3.06		0.166
LV mass index (g/m^2^)	172.16 ± 40.06	73.84 ± 15.85		0.002
GLSLV (%) ^2^	8.97 ± 2.94	17.71 ± 1.90		0.002
LA volume ^3^ (mL/m^2^)	58.15 ± 13.34	28.63 ± 3.92		0.003
RA volume ^4^ (mL/m^2^)	41.7 ± 17.55	19.74 ± 4.95		0.077
RV basal diameter ^5^ (mm)	37.4 ± 6.13	34.37 ± 5.25		0.405
PASP (mmHg) ^6^	44.88 ± 11.94	24.40 ± 2.53		0.017
RV area in diastole	17.98 ± 9.48	19.20 ± 5.83		0.913
FAC (%) ^7^	34.7 ± 9.46	46.56 ± 10.85		0.184
S Wave RV (m/s)	6.69 ± 2.87	11.17 ± 2.23		0.056
TAPSE ^8^	12.88 ± 4.97	20.89 ± 3.36		0.067

* NOTE: Values are expressed as absolute value (percentage), mean ± standard deviation. ** NOTE: Values are expressed as mean +/− standard deviation. Abbreviations: 1: LVEF—left ventricule ejection fraction, 2: GLSLV—global lontidudinal strain left ventricle, 3: LA—left atrium, 4: RA—right atrium, 5: RV—right ventricle, 6: PASP—pulmonary artery systolic pressure, 7: FAC—fracional area change, 8: TAPSE—tricuspid anular plane systolic excursion.

## Data Availability

Research data available at https://drive.google.com/drive/folders/1OTkVpsVm2WSZBmhRTR3VsdURKhRCD3za (accessed on 15 January 2025).

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
