# Peer review of "Different Gut Microbiome Profiles in Patients with Transthyretin Amyloidosis with and Without Cardiac Involvement"

_ijms, 2025, doi:10.3390/ijms26041689_

Round 1
Reviewer 1 Report
Comments and Suggestions for Authors
The authors present a study of the microbiome in patients with ATTR amyloidosis compared to controls. It is not surprising that the microbiota is different in these patients compared to controls, however, there is very limited research on this topic. The study is based on a small number of subjects, which is natural given the rareness of the disease, but the results are indeed of interest for the amyloidosis community. Still, I have some comments and suggestions for revision before the manuscript is ready for publication.
1. Please follow the latest nomenclature guidelines from the International Society of Amyloidosis throughout the manuscript.
2. The Methods section seems to be misplaced in the end of the manuscript, please consider revising.
3. Results: The study includes both ATTRv and ATTRwt patients. Since the phenotype differs somewhat between the genotypes it would be interesting to separate these in the analyses. ATTRwt patients usually suffer mainly from cardiomyopathy and not from polyneuropathy, whereas the opposite can be true for ATTRv patients. If separate analyses between genotypes are not possible please note in Table 1 how the genotypes were distributed between the groups and at least mention how this potentially could influence the results.
4. Results, Table 1: The age and sex is significantly different between the groups. I understand that this might be difficult to adjust, but this may have influenced the results. Sex was mentioned in the Discussion but not age, please consider updating.
5. Diabetes, Table 1: Like age and sex, the frequency of diabetes differs significantly between the groups with no diabetics in group 2. Since diabetes may cause GI dusturbances and changes in the gut microbiota, this is even more important. Best would be to exclude the diabetics from the analyses but otherwise this must at least be highlighted in the Discussion.
6. Figs 4 and 5 are a bit hard to follow, please update figure and/or legends to make them easier to understand. For example, what does the change on the x axis stand for?
7. Discussion, Limitations: The ATTRh form is denoted "wild type", which is not correct. Should it be "variant" instead?
8. Discussion: Please add some words on what you think are the main reasons for the differences in microbiota between the studied groups. Is it mainly due to GI disturbances in ATTR amylodiosis or perhaps to the cardiac effects? Since ATTR amylodosis is a multi-systemic disease, there is much that can contribute to the findings but I would be interested to hear what the authors think are the most important factors in the different groups.
Comments on the Quality of English LanguageThere are a few misspelled words and some sentences that are hard to follow. Please consider revising.
Reviewer 2 Report
Comments and Suggestions for Authors
Authors analyzed the potential association of intestinal and cardiac amyloidosis by comparison of data from blood, gut and heart. They suggest an interesting connection and potential mediators of both alterations sharing potential common mechanisms. The number of patients is scarce, but the disorder is also infrequent. The methodology must be improved, and other issues must be repaired before being published:
- The diet of patients can be essential for bacteria distribution and function in gut, and thus, it may interfere with enterocytes pathophysiology. Also, it may influence on gut leaking and blood content of bacterial metabolites
- Lifestyle, physical exercise and smoking habits also should have been collected form subjects
- How and when were the fecal samples taken and stored?
- On exclusion criteria, did you include gut allergies or other diseases (colon cancer)?
- Did you test the association between plasma biomarkers and heart failure? And among bacteria metabolites and plasma biomarkers? Did you measure glucose and the lipid profile?
- Does any mutation associate with gut or cardiac alteration? And with plasma mediators?
- How did you “confirm the TTR protein in the tissue”?
Round 2
Reviewer 2 Report
Comments and Suggestions for Authors
Dear Authors, some concerns need to be added or explained:
- “The diet of patients can be essential for bacteria distribution and function in
gut, and thus, it may interfere with enterocytes pathophysiology. Also, it may
influence on gut leaking and blood content of bacterial metabolites.”
R_ I do agree. We attempt to mitigate this potential central bias by choosing
controls who used to live with the patients and share similar eating habits.
Please refer to the sentence in green on page 2.
This is not enogugh and must be considered and included as a truly limitation of the study
“Lifestyle, physical exercise and smoking habits also should have been
collected form subjects.”
R_ I agree. From the beginning, it would have been very beneficial to collect these
features in the analysis. Still, I believe somehow, partially, this data may be
portrayed by the performance on the sixty-minute walking test.
Please refer to the last topic in green on page 3.
This is not enogugh and must be considered and included as a truly limitation of the study
“On exclusion criteria, did you include gut allergies or other diseases (colon
cancer)?”
R_ These features were not pointed out in the exclusion criteria, but considering
the small number of participants, I can assure you there were no significant
allergies or advanced cancer as concomitant diseases during the study period.
This is not enogugh and must be considered and included as a truly limitation of the study. The allergies could have been diagnosed even before recruitment
“Did you test the association between plasma biomarkers and heart failure?
And among bacteria metabolites and plasma biomarkers? Did you measure
glucose and the lipid profile?”
R_ Due to the limited budget, we had to focus on the aspects already portrayed.
But yes, including these data would have been very beneficial for the analysis.
This is not enogugh and must be considered and included as a truly limitation of the study. Biomarkers for heart failure as well as bacterinemia should have been included. At least, statistic analysis should be done with the markers you got
“Does any mutation associate with gut or cardiac alteration? And with plasma
mediators?”
R_ Yes. The most common mutation was ATTRV142I (52.9% among ATTRv
patients). Please see the sentence in green on page 13. Concerning the plasma
mediators, I do not have this data.
An statistic analysis should be done to test these associations
“How did you “confirm the TTR protein in the tissue?”
R_ I had also wondered about that issue until I came across the research
conducted by Dr. Gillmore in London (2016). His team demonstrated that, in the
absence of circulating immunoglobulins, scintigraphy is enough to confirm the
diagnosis without needing biopsies. https://pubmed.ncbi.nlm.nih.gov/27143678/
I highlighted it in green on page 3.
You should state what you made; no what other researcher says
Round 3
Reviewer 2 Report
Comments and Suggestions for Authors
Some issue must be done:
“Did you test the association between plasma biomarkers and heart failure?
And among bacteria metabolites and plasma biomarkers?
Biomarkers for heart failure as well as bacterinemia should have been included. At least, statistic analysis should be done with the markers you got
“Does any mutation associate with gut or cardiac alteration? And with plasma
mediators?”
More details are needed for these new approaches: R_ Scintigraphy with technetium pyrophosphate indicating levels 2 or 3 of uptake confirmed the TTR protein in the tissue.
